# Post-Concussion Syndrome and Sleep Apnea: A Retrospective Study

**DOI:** 10.3390/jcm9030691

**Published:** 2020-03-04

**Authors:** Alexandra Santos, Hannah Walsh, Neda Anssari, Ivone Ferreira, Maria Carmela Tartaglia

**Affiliations:** 1Tanz Center for Research in Neurodegenerative Diseases, University of Toronto, Toronto, ON M5S 1A8, Canada; alexc.santos@mail.utoronto.ca (A.S.); hannah.walsh94@gmail.com (H.W.); 2Toronto Western Hospital, Division of Neurology, Toronto, ON M5T 2S8, Canada; neda.anssari@uhn.ca; 3Canadian Concussion Centre, Krembil Neuroscience Center, Toronto, ON M5T 2S8, Canada; 4Toronto Western Hospital, Asthma and Airway Centre, Toronto, ON M5T 2S8, Canada; ivone@drferreira.ca; 5Department of Medicine, McMaster University, Hamilton, ON L8S 4L8, Canada

**Keywords:** concussion, post-concussion syndrome, sleep apnea, cognition, persistent symptoms

## Abstract

Background: Concussion symptoms typically resolve within 7–10 days, but 10–25% of patients do not fully recover. They can develop post-concussion syndrome (PCS), which includes sleep abnormalities such as obstructive sleep apnea. It is unclear how specific sleep problems manifest in PCS and how it relates to cognition and symptomology. Methods: A retrospective chart review was conducted on PCS patients seen at the University Health Network (UHN) Concussion Clinic and sent for sleep study. Neuropsychology tests, concussion features, PCS symptoms, and demographics were abstracted from clinical charts. Sleep measures were abstracted from the overnight sleep study. Data were analyzed using chi-squared tests and linear regression. Results: Fifty-one patients completed the sleep study; 78% of these were diagnosed with sleep apnea. Patients with sleep apnea reported significantly more memory symptoms. A trend existed for higher total symptom number. Age was significantly different between the two groups. Women and men were equally at risk of being diagnosed with sleep apnea. Conclusions: Sleep apnea is common in PCS patients complaining of non-restorative sleep and/or waking up with headaches. Sleep apnea was associated with more memory symptoms. PCS patients are at higher risk for sleep apnea and sleep study should be considered if complaining of non-restorative sleep and/or waking up with headaches, regardless of sex and other known risk factors.

## 1. Introduction

Concussion or mild traumatic brain injury (mTBI) is the most common form of traumatic brain injury [1]. Most concussion symptoms resolve within 7–10 days, but 10–25% of patients do not fully recover and have persistent symptoms one month after the injury [2]. This persistence of symptoms is referred to as post-concussion syndrome (PCS) or persistent symptoms of concussion [3,4]. Patients with PCS can have a variety of symptoms including physical symptoms (e.g., headaches, dizziness, difficulty with balance, tinnitus), cognitive symptoms (e.g., memory impairment, executive dysfunction, attention deficits), and emotional symptoms (e.g., depression, anxiety) [1,4]. Sleep abnormalities such as insomnia, hypersomnia, and sleep apnea [5] are also frequently observed in PCS patients [6,7].

One of the sleep disturbances reported in patients with PCS is sleep apnea [4]. Sleep apnea is characterized as repeated interruptions of breathing during sleep [8] and is classified into two general categories: obstructive, which is caused by an upper airway collapse or a physical blockage [8], and central, which is caused by dysfunction of the physiological centers for breathing rhythm [8]. The prevalence of sleep apnea across all severities of traumatic brain injury is 25% [5], whereas that of the general adult population is 6.4% from the 2016 and 2017 Statistics Canada report [9]. Risk factors in the general population are being overweight (higher body mass index (BMI)) [9], older age (prevalence three times higher in peopled aged 60–79 years [9] compared to 59 years and below), and male sex (three times higher risk than women) [10].

Sleep apnea is associated with a number of negative health and functional consequences [8,11]. This includes cognitive deficits [12], impaired neural repair via reduced synaptogenesis [8,13], and increased likelihood of motor vehicle or workplace accidents [8]. Some possible explanations for this increased incidence of obstructive sleep apnea (OSA) in patients with PCS may be medications used for symptomatic treatment of concussion like sedatives, hypnotics, and opioids as well as increased weight gain that commonly occurs after concussion [8]. PCS patients are already at risk of cognitive impairment [1], and comorbid sleep-disordered breathing could potentially aggravate their symptoms. Previous studies showed that sleep difficulty is predictive of prolonged recovery from concussion symptoms and increased severity of sleep disturbance is associated with the severity of post-concussive symptoms [14,15]. Overall, undiagnosed and untreated sleep apnea in PCS patients can have a negative impact on their recovery.

There is limited literature on the sleep abnormalities specifically in PCS; most of the relevant sleep disturbance literature pertains to concussion in the acute phase [5] (not necessarily with persistent symptoms) or groups all severities of brain injuries together rather than investigating concussion separately [8]. Due to this gap in the PCS sleep literature, it is unclear how prevalent specific sleep problems are in PCS, how they manifest, and how it may be related to PCS patients’ memory, cognition, and other symptoms.

In this retrospective study, we assessed the number of patients with PCS seen in a two year time period at the University Health Network (UHN) Concussion Clinic, that completed a sleep study and had sleep apnea, the factors associated with sleep apnea in this population, as well as the relationship between sleep apnea, concussion symptoms and features, and cognitive measures. We hypothesized that PCS patients complaining of non-restorative sleep and/or waking up with headaches will likely have sleep apnea, and that sleep apnea is related to poor performance on neuropsychological testing and more post-concussion symptoms.

## 2. Materials and Methods

This was a retrospective chart review of patients diagnosed with PCS by a cognitive neurologist at the University Health Network Concussion Clinic, a tertiary care center. Patients with persistent symptoms after a concussion are referred by family doctors or other physicians. Patients are diagnosed with PCS based on modified DSM-IV criteria [16], which include (1) history of head trauma that causes a significant cerebral concussion, and (2) at least three or more of the following symptoms persisting for three or more months: being fatigued easily; disordered sleep; headache; vertigo or dizziness; irritability or aggression on little or no provocation; anxiety, depression, or affective lability; apathy or lack of spontaneity; and other changes in personality (e.g., social or sexual inappropriateness). We accepted that subjective or objective impairments in memory or attention and neuropsychological testing could be normal. This study was approved by the University Health Network Research Ethics Board.

### 2.1. Participant Inclusion and Exclusion

The inclusion criteria for this study were (1) at least one visit to the UHN Concussion Clinic between 10 January 2017 to 5 March 2019; (2) PCS diagnosis; and (3) recommendation to complete a sleep study. The patient was excluded if they (1) had severe dementia, (2) had a prior sleep apnea diagnosis before coming to the clinic and only had continuous positive airway pressure (CPAP) titration available, (3) had failed the sleep study, or (4) we could not obtain their sleep study results.

### 2.2. Chart Data

Data were abstracted using both the Electronic Patient Records system at UHN and manual chart review. From the patients included for the analysis, data were taken from their concussion physician interview document. The following variables were collected: date of last concussion, concussion mechanism, number of concussions, time since last concussion, number of memory function symptoms from a maximum of five (i.e., repeating oneself, misplacing objects, forgetting appointments, leaving appliances on, remote memory deficit), executive function symptoms from a maximum of eight (i.e., difficulty planning/organizing, decreased concentration/attention span, difficulty multitasking, poor judgement and decision-making, difficulty problem solving, restlessness, mental rigidity/inflexibility, repetitive behaviors), number of personality changes from a maximum of 13 (i.e., aggression, agitation, apathy, disinhibition, impulsivity, loss of social graces, loss of empathy, emotional lability, obsessive/compulsive behaviors, criminal/violent behaviors, hygiene, eating habits, ritualistic behaviors), number of neuropsychiatric symptoms from a maximum of seven (i.e., low mood/sadness, feelings of failure, anxiety, irritability, suicidal thoughts, hallucinations/delusions, appetite), sleep-related symptoms from a maximum of five (i.e., excessive daytime sleepiness, restless legs, shortness of breath at night/snoring, trouble getting to sleep, trouble maintaining sleep), and number of constitutional symptoms from a maximum of 10 (i.e., headaches, neck pain/tightness, vertigo or dizziness, double or blurred vision, photophobia, phonophobia, nausea or vomiting, fatigue, numbness or paresthesia, hearing problems). Date of birth and sex were also obtained.

### 2.3. Cognitive and Memory Performance

Neuropsychological measures were obtained from raw Toronto Cognitive Assessment (TorCA) [17] domain scores that the patient completed on their initial consultation at the UHN Concussion Clinic. The TorCA is a comprehensive neuropsychological test with 27 subtests designed to detect cognitive impairment. It tests seven cognitive domains including orientation, immediate recall, delayed recall, delayed recognition, visuospatial function, working memory/attention/executive control, and language [17].

The four domains analyzed in the present study were immediate recall, delayed recall, delayed recognition, and working memory/attention/executive control. Immediate recall was determined by calculating the mean of the three recall trials of the Consortium to Establish a Registry for Alzheimer’s Disease (CERAD) 10-word list [18]. Delayed recall was determined by summing the 10 min delayed recall of the CERAD word list and the Benson figure [19]. Delayed recognition was determined by summing the recognition of the CERAD words score and the score of Benson figure recognition. Working memory and attention were determined by summing the Digit Span and serial subtractions. Executive control was determined by summing the drawing alternating sequences, verbal letter fluency, and Trail making A and B [20].

In addition to the four domains and total scores, Digit Span Forward, Digit Span Backward, and Trails B z-scores were analyzed separately. The Trails B z-score was calculated with the formula, z = (x – μ)/σ, using published normative data, where x is the patient’s raw Trails B score, μ is the published mean Trails B score for the patient’s demographic group, and σ is the published standard deviation for the patient’s demographic group [21].

### 2.4. Sleep Data

The sleep data were collected from the patient’s polysomnogram, completed in an overnight diagnostic sleep study. The sleep variables included: body mass index (BMI), sleep stage architecture, including rapid eye movement (REM) stage sleep, and non-REM stage 3 sleep, i.e., slow-wave sleep (% of total sleep time (TST)), sleep efficiency (TST/Time in bed), Epworth Daytime Sleepiness Scores, apnea-hypoapnea index (AHI), and sleep apnea diagnosis. Sleep apnea diagnosis was determined using the following AHI thresholds: Mild: 5 ≥ AHI < 15, Moderate: 15 ≥ AHI < 30, Severe: AHI ≥ 30 [22]. The Epworth Daytime Sleepiness scores were not analyzed further as over 25% of the patients did not have a score available.

### 2.5. Statistical Analysis

The data were analyzed using R software. Data were compared between the PCS patients with sleep apnea with those without sleep apnea using mainly descriptive and comparative statistics.

Percentages were used to illustrate the proportion of PCS patients included for the study and the patients who also completed the sleep study. The PCS patients with or without sleep apnea were compared in terms of their demographics. A Fisher exact test was used to evaluate if there was a significant difference in the differences in sex distribution, concussion number, and injury mechanism between PCS patients with and without sleep apnea. Within the PCS patients with sleep apnea, the number and percentage of apnea severity (mild, moderate, or severe) was calculated.

A Mann–Whitney non-parametric test was used to compare the PCS patients with and without sleep apnea for the following parameters: time since last concussion, age, and BMI. A multivariate linear regression was used to compare the PCS patients with and without sleep apnea for the following parameters: number of memory symptoms, number of executive symptoms, number of personality symptoms, number of psychiatric symptoms, number of constitutional symptoms, total symptoms, immediate recall scores, delayed recall scores, delayed recognition scores, working memory/attention/executive control scores, Digit Span Forward, Digit Span Backward, Trails B z-score, and overall TorCA scores. Since age was significantly different between the groups, it was added along with sleep apnea as an explanatory variable in the linear regression, except for when analyzing the Trails B z-score as age was already accounted for in the z-scores. The same was conducted for the objective sleep parameters: REM sleep, slow-wave sleep, and sleep efficiency. Bonferroni correction was used to correct for multiple comparisons within each category of variables.

## 3. Results

### 3.1. Study Population

There were 532 patient visits in the UHN Concussion Clinic between 10 January 2017 and 5 March 2019; see Figure 1 for flow of patient selection. Of these appointments, there were 355 different patients seen, i.e., many patients were seen more than once during this time period. From the 355 concussion clinic patients, 107 (30.1%) were recommended to complete a sleep study based on clinician discretion (MCT) that included (1) the patient complaining of non-restorative sleep and/or (2) waking up with headaches. There were 22 patients who refused to complete the sleep study and 29 patients in which the status of their sleep study was unknown based on their clinical chart, and thus no sleep data were available. The remaining 56 (52.3%) patients completed the overnight sleep study. There were five patients excluded and the remaining 51 patients were included for analysis.

### 3.2. Demographics and Concussion Features

From the 51 patients included for the analysis, 40 (78.4%) were diagnosed with OSA and 11 (21.6%) did not have sleep apnea. These two groups were compared to investigate the impact of sleep apnea. Amongst the 40 sleep apnea patients, 17 (42.5%) had mild OSA, 13 (32.5%) had moderate OSA, and 10 (25.0%) had severe OSA. If all patients referred for sleep study are considered (n = 107), the prevalence is 40/107 (37.3%).

Table 1 contains demographic information for the PCS patients with and without sleep apnea. Age was significantly different between the two groups with PCS patients with sleep apnea being significantly older than those without (53.55 ± 13.23 vs 36.6 ± 12.17 years, *p* = 0.002). Although their mean age was older, 29/40 (72.0%) of the PCS patients in the sleep apnea group were under 65 years of age in the sleep apnea group. In comparing time between last concussion and the overnight sleep study, it was significantly longer in PCS patients with sleep apnea (*p* = 0.01).

There were no significant differences between the PCS patients with and without sleep apnea with respect to sex distribution, concussion number, and mechanism of injury with the most to the least common mechanism being motor vehicle accidents, trauma, and sports in both groups. Multiple mechanisms were the least common and only present in the sleep apnea group.

### 3.3. Neuropsychology Performance and PCS Symptoms

There was a significant difference (*p* = 0.02) after correction for multiple comparisons between the number of memory symptoms of PCS patients with sleep apnea compared to those without (Table 2). The PCS patients with sleep apnea had more symptoms in each domain and total than the PCS patients without sleep apnea, but the differences were not significant.

There was a significant difference (*p* = 0.04) before correction for multiple comparison between immediate recall scores of PCS patients with sleep apnea and those without (Table 2). Although not significant, the mean scores of the neuropsychology scores for the PCS patients with sleep apnea were always slightly lower than the scores of the patients without sleep apnea.

There was no significant difference between sleep parameters, REM sleep, slow-wave sleep, and sleep efficiency between PCS patients with and without sleep apnea.

## 4. Discussion

Our results demonstrate that a majority of people with PCS who were sent for and completed a sleep study were diagnosed with some form of sleep apnea, The sleep apnea prevalence across all brain injury severities is 25% [7] and our estimated value of 11.3% in the PCS population indicates a lower prevalence, but still higher than the general population (6.4%) [9]. There were many patients that were whose sleep study status remained undetermined or they did not complete an overnight sleep study after the physician’s referral, thus the real prevalence of sleep apnea in this PCS cohort could not be calculated. Although the reasons patients did not complete the study is multifactorial, some potential reasons include inability to find overnight childcare, no inability to get a sleep clinic nearby, and a disinterest in doing a sleep study.

There were no significant differences between the PCS patients with versus without sleep apnea in concussion features like mechanism and number; thus, these concussion factors do not seem to play a major role in the risk of sleep apnea in PCS.

A significant risk factor for sleep apnea in this study was increased age. This is consistent with sleep apnea risk factors in the general population [22]; however, the risk for sleep apnea is typically higher in individuals aged 65 years or older [22] and the mean age in our study population was only 53.6 years old. Of the 40 PCS patients with sleep apnea, 29 (72%) were under 65 years of age. These findings indicate that although older age is a risk factor for sleep apnea in the PCS population, younger PCS patients who have symptoms suggestive of sleep apnea should also be assessed. Another study looking at the incidence of sleep apnea in traumatic brain injury and also found that age was the most significant risk factor [8].

A notable result in this study is that sleep apnea was equally prevalent amongst women and men, whereas in the general population, men are three times more likely to be diagnosed with sleep apnea [10]. These results underscore the necessity for enquiring about symptoms suggestive of sleep apnea in women with PCS.

A high BMI is a known risk factor for sleep apnea [9], but our study found no significant difference in BMI between the PCS patients that were diagnosed with sleep apnea and those without. Although both groups had high BMI, our results suggest that additional factors in addition to BMI contribute to PCS patients’ risk of sleep apnea. Our results concur with a previous study that reported a much higher prevalence (36%) of sleep apnea in the traumatic brain injuries of all severities than the general population [23] but also did not find a significant relationship with BMI.

Time between last concussion and the overnight sleep study was significantly higher in patients with sleep apnea. This may be partially explained by this group being older or may be related to untreated sleep apnea contributing to persistence of post-concussion symptoms.

### 4.1. Sleep Apnea and Sleep Architecture in PCS

There were no significant differences between PCS patients with sleep apnea and those without in the objective sleep measures from the overnight sleep study. There were no controls in our study, so it is difficult to determine whether there were significant changes in sleep architecture across all PCS patients.

### 4.2. Sleep Apnea, PCS Symptoms, and Neuropsychology Performance

Sleep-disordered breathing has been associated with cognitive impairment across many domains including attention/working memory, global cognition, and delayed recall [12]. Similarly, the present study revealed significantly lower immediate recall scores in PCS patients with sleep apnea compared with those without, although it did not survive multiple comparisons. PCS patients with sleep apnea reported significantly more memory symptoms than those without sleep apnea. Additionally, although not significant, the PCS patients with sleep apnea had slightly more symptoms in each domain and more total symptoms than the PCS patients without sleep apnea. Since concussion features did not differ between the two groups, the increased symptoms may be a result of sleep apnea. Furthermore, it has been demonstrated that sleep disorders contribute to cognitive impairment in all severities of TBI [24].

The study patients presented with a number of hallmark PCS symptoms including complaints about mood and headache. Many PCS symptoms such as pain, anxiety, and depression are also seen in secondary sleep disturbances [4]. Since a concussion is associated with cognitive symptoms such as forgetfulness [4] and slowed thinking [7], as well as a multitude of somatic and mood issues [1], the full impact of sleep apnea may have been obscured in the current small study. A prospective study measuring the change in symptoms as well as cognitive performance and mood before and after treatment of sleep apnea would be necessary to further elucidate the relationship between sleep apnea and concussion.

The means through which sleep apnea could contribute to PCS symptoms are likely multifactorial. The hippocampus and frontal cortex are particularly vulnerable to sleep fragmentation, hypoxemia, and hypercapnia, and are closely associated with memory processes and executive functions [25]. There are a number of studies demonstrating altered brain metabolism and neuronal loss in these areas in OSA patients [26,27,28] including axonal dysfunction or loss in the frontal white matter [26]. Reduced cortical gray matter volumes, as well as the size of the right hippocampus and right and left caudate, were smaller in OSA patients compared to the control group and correlated with poorer cognitive performance [29].

### 4.3. Strengths, Limitations, and Future Work

Although a retrospective study, the objective measure of sleep with polysomnography in 51 patients lends support to the results that OSA is common in PCS patients. In addition, these were well-characterized patients using neuropsychological testing, so the results warrant further investigation with a treatment trial to evaluate the impact of treatment of sleep apnea on PCS-related symptoms.

One limitation to this study was the small sample size that precluded analysis of the impact of medications and specific comorbidities. Additionally, this study was based on retrospective data and so lacks sex- and age-matched controls. There were no controls in our study, so was is difficult to determine whether the lack of difference in sleep architecture between PCS patients with and without sleep apnea cannot exclude changes across all PCS patients irrespective of sleep apnea. A limitation with overnight sleep studies is that they may not truly reflect the sleep the patient usually gets at home; however, they are currently the gold standard for sleep research.

The high number of sleep apnea diagnoses in the present study suggests that people with PCS are at high risk of sleep apnea and should routinely be investigated further for sleep-disordered breathing. The study highlights that women with PCS should be investigated when they have sleep complaints or awaken with headaches and high BMI as they seem to be at the same risk as men for sleep-disordered breathing. Also, although age increased the risk of a diagnosis of sleep-disordered breathing there were many younger people that were also diagnosed, suggesting that the typical OSA profile may not apply to people with PCS.

The results of this study indicate that many patients with PCS and sleep complaints are at risk for sleep apnea. The overlap between sleep apnea and PCS symptoms is considerable and so warrants a prospective treatment trial to address whether interventional studies of sleep apnea could ameliorate PCS symptoms in patients with PCS and sleep apnea.

## Figures and Tables

**Figure 1 jcm-09-00691-f001:**
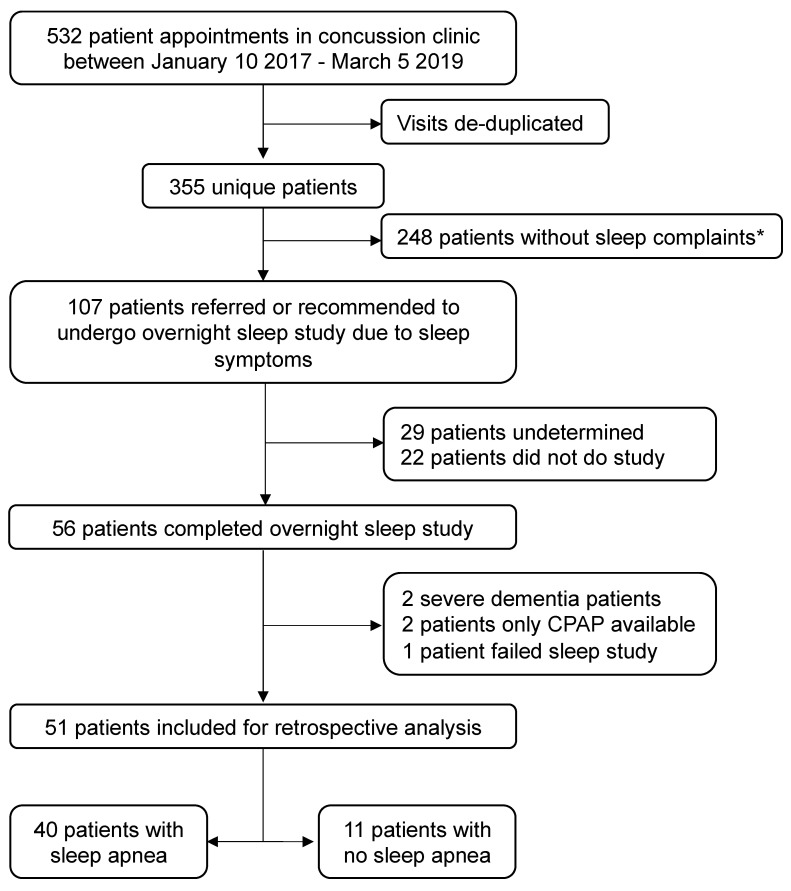
Flow of patient selection and exclusion. * Sleep complaints are (1) the patient complaining of non-restorative sleep and/or (2) waking up with headaches.

**Table 1 jcm-09-00691-t001:** Demographics and concussion features of all study patients.

	Apnea Patients, n (%)	No Apnea Patients, n (%)	*p*-Value
Total (n)	40	11	
Female	17 (42.5)	6 (54.5)	0.514
Male	23 (57.5)	5 (45.5)	
1 Concussion	20 (50)	6 (54.5)	1
2+ Concussions	20 (50)	5 (45.5)	
Multiple (Mechanisms)	4 (10)	0 (0)	0.696
Motor Vehicle Accident	17 (42.5)	7 (63.6)	
Sports	5 (12.5)	1 (9.1)	
Trauma	14 (35)	3 (27.3)	
	Mean ± SD	Mean ± SD	
Time Between Last Concussion and Overnight Sleep Study (years)	4.42 ± 7.21	1.1 ± 0.94	0.013
Time Between Last Concussion and Neuropsychological Testing (years)	3.71 ± 7.2	1.0 ± 1.32	0.369
Age	53.55 ± 13.23	36.6 ± 12.17	0.002
Body Mass Index (BMI)	29.9 ± 4.21	26.49 ± 7.27	0.102

**Table 2 jcm-09-00691-t002:** Neuropsychology test scores and post-concussion syndrome (PCS) symptoms of all study patients.

Neuropsychological Test/Symptom Report	Apnea Patients (Mean ± SD)	No Apnea Patients (Mean ± SD)	*p*-Value	Corrected *p*-Value
Immediate Recall Score (/30)	18.1 ± 4.03	22.2 ± 3.49	0.048	0.383
Delayed Recall Score (/27)	18 ± 4.48	19.5 ± 3.21	0.515	1
Delayed Recognition Score (/21)	19.49 ± 1.76	20.6 ± 0.7	0.136	1
Working Memory/Attention/Executive Control Score (/123)	103.6 ± 12.51	105.2 ± 9.96	0.727	1
Trails B z-Score	1.34 ± 2.5	1.86 ± 4.21	0.616	1
Digit Span Forward (/9)	6.47 ± 1.6	6.7 ± 1.25	0.82	1
Digit Span Backward (/8)	4.03 ± 1.53	4.9 ± 1.91	0.45	1
Overall Score (%)	85.41 ± 7.43	87.48 ± 6.38	0.925	1
Symptom Domain				
Memory (/5)	3.12 ± 1.11	1.91 ± 1.58	0.003	0.022
Executive (/8)	4.45 ± 2.22	3.36 ± 2.69	0.025	0.173
Personality (/13)	2.6 ± 1.86	1.82 ± 1.94	0.083	0.579
Sleep (/5)	3.5 ± 1.5	3 ± 1.48	0.237	1
Neuropsychiatric (/7)	2.27 ± 0.88	1.91 ± 1.58	0.401	1
Constitutional (/10)	4.9 ± 2.71	4.64 ± 1.75	0.272	1
Total (/48)	18.57 ± 6.22	14.73 ± 6.68	0.007	0.051

Neuropsychology scores are from initial Toronto Cognitive Assessment (TorCA) testing. Symptom domain scores are from the concussion physician interview document.

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
