# Peer review of "Post-Concussion Syndrome and Sleep Apnea: A Retrospective Study"

_jcm, 2020, doi:10.3390/jcm9030691_

Round 1

Reviewer 1 Report

General: Sleep apnea (SA) could further impair cognition among individuals with post-concussive symptoms (PCS). The authors examined the prevalence of SA in a selected sample of individuals who had been diagnosed with PCS at a visit to a concussion clinic and assessed associations with baseline factors. While the authors highlight some interesting differences in scores on a variety of neuropsychological batteries, this study has significant flaws that limit interpretation of results. Furthermore, study conclusions aren’t supported by the results.

Introduction

The authors hypothesis that people with SA would have more PCS was not mentioned in the methods or tested. The first part of the hypothesis – that PCS patients would have a high prevalence of SA, is impossible to test with the chosen study design (i.e. the study sample is not representative of the target population AND not all patients with PCS were tested).

Methods and Results

Very little information is provided about the study site. Who comes to the concussion clinic? Is it people who are suffering from PCS? How is concussion evaluated? Are patients referred to the clinic from elsewhere? These questions help the reader understand who the study population is. How exactly was PCD diagnosed? Please mention the symptoms. What exactly was clinician discretion in terms of sending for a sleep study? This seems to be a pretty subjective statement. Were there people who complained of a headache that weren’t sent for a study? Were there other criteria that were evaluated? Only 14% of the entire sample completed a sleep study – this is such a small percentage that we can’t draw any conclusions about whether these 51 people represented the entire sample, especially since no information was provided on those who didn’t complete the study. The statistical methods used aren’t appropriate. Cell sizes under 5 suggest that Fishers’ exact test should have been used for some categorical variables. Linear regression was an odd choice for testing differences between the exposed groups, especially because it seems that the authors may have controlled for age for some variables in Table 1 but not others. Furthermore, the skewed ‘time since last concussion’ should have probably been examined using a non-parametric method. Use of a Bonferroni correction in Table 1 was unnecessary. Overall, the impression I have is that the statistical analysis lacked rigor. We have no information on how long the neurpsych testing was performed after concussion.

Discussion

The overall conclusion that there is a high prevalence of SA among individuals with PCS tested for SA doesn’t seem to be based on results from this study in which 40/355 (11%) individuals were diagnosed with SA.

Reviewer 2 Report

This is a well written manuscript. It is well designed, well described, and not overstated. This study has relevance to the field and relevance to clinical practice.  

It is notable that the threshold for ordering a sleep study was quite low--1) non-restorative sleep and/or 2) waking up with headaches--and yet amongst this group, rates of OSA were still so high. 

You reference a few studies that similarly have found high rates of OSA after TBI. However, I would like to see a line or two regarding mechanisms--why such a high rate of OSA after TBI, how does this impact cognitive functioning (memory consolidation, etc?) and by interfering with "neural repair mechanisms" what are you referring to? 
